# A Cultural Conscience for Conservation

**DOI:** 10.3390/ani7070052

**Published:** 2017-07-20

**Authors:** Caroline Good, Dawn Burnham, David W. Macdonald

**Affiliations:** Wildlife Conservation Research Unit, University of Oxford, Oxford OX13 5QL, UK; david.macdonald@zoo.ox.ac.uk (D.W.M.); dawn.burnham@zoo.ox.ac.uk (D.B.)

**Keywords:** culture, species royalty, funding, sports mascots, fashion animals, national animals, heritage

## Abstract

**Simple Summary:**

This opinion piece explores how implementing a species royalty for the use of animal symbolism in affluent cultural economies could revolutionise conservation funding. A revenue revolution of this scale is urgently necessary to confront the sixth mass extinction that the planet is now facing. But such a revolution can only occur if the approach to conservation now evolves quickly across disciplines, continents, cultures and economies. This piece is a call to action for research-, culture-, and business-communities to implement a new ethical phase in economic policy that recognises the global cultural debt to the world’s most charismatic wildlife species.

**Abstract:**

On 2 July 2015, the killing of a lion nicknamed “Cecil” prompted the largest global reaction in the history of wildlife conservation. In response to this, it is propitious to consider the ways in which this moment can be developed into a financial movement to transform the conservation of species such as the lion that hold cultural significance and sentiment but whose numbers in the wild are dwindling dangerously. This provocative piece explores how a species royalty could be used effectively by drawing revenue from the heavy symbolic use of charismatic animals in affluent economies. This would, in turn, reduce strain on limited government funds in threatened animals’ native homelands. Three potential areas of lucrative animal symbolism—fashion, sports mascots, and national animals—provide examples of the kind of revenue that could be created from a species royalty. These examples also demonstrate how this royalty could prove to be a desirable means by which both corporations and consumers could positively develop their desired selves while simultaneously contributing to a relevant and urgent cause. These examples intend to ignite a multi-disciplinary conversation on the global cultural economy’s use of endangered species symbols. An overhaul in perspective and practice is needed because time is running out for much of the wildlife and their ecosystems that embellish products and embody anthropocentric business identities.

## 1. Introduction

Could a cultural conscience play a part in conserving species that abound in symbolic imagery, but whose numbers are rapidly declining in the wild? When a model, a design, or a piece of music is used for the promotion of something other than itself, a fee is charged for that privilege. What if each time the symbol of an endangered animal was used the species, or the effort to conserve it, was paid a royalty? This principle, applied to threatened, charismatic animals [1], could revolutionise funding for conservation; and just such a revolution is needed to reverse the current tumble to extinction. 

Here, we consider the financial potential of a cultural conscience with respect to two affluent, global markets—fashion and sport—that crawl with animal imagery, as well as the cultural relationship of economically buoyant Great Britain with its threatened non-native national animal—the lion. Exploring these three symbolic arenas in retrospect, they are intended to highlight just some of the possibilities for translating animal imagery into conservation revenue for the future.

## 2. The Lion′s Share

The lion is the most frequently invoked national animal, adopted by 15 countries worldwide from Singapore to Sri Lanka, Libya to Luxemburg. Today, the lion is extant in only two of these nations, Ethiopia and Kenya. But that has not deterred its symbolic, cultural proliferation. The British lion, introduced in the 13th century by King Richard “the Lionheart”, arrived millennia after lions themselves strolled through Pleistocene England. Richard I′s Plantagenet lions remain to this day. Over the centuries, they have been placed on shields to scare the enemy, featured on flags and in crests for identification at sea and abroad, and posted in marble and stone as guardians of public buildings and national treasures. In more recent years, the British lion symbol has migrated onto food as a stamp of quality and as emblems of the country′s national sports teams, from rugby and football to Olympic team GB (Figure 1).

But despite the lion′s evident cultural appeal, its global population has fallen by approximately 43% over the past 21 years (and by 65% in West and Central Africa) [2]. How might the lion′s symbol, so familiar as a marketing instrument, metamorphose once again, this time into much needed conservation revenue?

Take eggs. According to the British Egg Industry Council, 34 million eggs are consumed in Britain each day. The British Lion Quality Seal is stamped on approximately 85% of these. If each lion stamp were to earn the species one tenth of a penny, then every day it would receive £28,900. That is £10.5 million a year.

A few shelves over in Britain′s supermarket baking aisles sits Lyle′s Golden Syrup. The tins′ Old Testament centrepiece—Samson′s riddle “What is sweeter than honey? What is stronger than a lion?” [3]—is evoked by a lion corpse swarming with bees. The picture, and its accompanying rubric “Out of the strong came forth sweetness”, all allude to the superior sweet product inside being the answer to Samson′s puzzle. While Lyle′s Golden Syrup range has now extended to cakes and easy-squeezy bottles, its packaging has remained unchanged since the product′s inception in 1883. The equivalent of 40 million tins of the syrup are produced every year. If 10 pence per tin were split between bee and lion conservation, that would be £2 million a year to the lion.

Out of the supermarket and onto the sports field, the Premier League′s crowned lion logo is the hallmark of its merchandise. The League′s clubs are forecast to sell over five million shirts collectively by the end of this season [4]. If the Premier League donates one pound for every shirt sold, over 4000 lion guardians could be employed for one year [5] (This calculation is based on 5 million shirts being sold per year, with £1 per shirt being donated to make a combined total of £5 million. One lion guardian costs £1410 ($1800) per annum, allowing for the £5 million to cover 4000 lion guardians over one year. A Lion Guardian is a trained individual who actively protects lions. The programme is currently active in East Africa and at present covers approximately 5500 sq. ks (1.3 million acres)).

The widespread and varied use of the lion in British culture is conducive to the idea that the inherent value of a national animal to a country’s identity potentially makes them a relatively easy target to garner public support for their protection [6]. In the twentieth century, the North American success stories of the bald eagle, *Haliaetus leucocephalus*, and the bison, *Bison bison*, exemplify how strong national and international protection—generated through both public and political support—have resulted in populations rebounding to healthy and sustainable levels [7]. In the United States, the endemic bald eagle′s image appears on passports, national defense, and all forms of legislation. Once threatened with extinction, they are now classified by the IUCN as being of Least Concern with their populations continuing to increase [8]. In the case of the bison, in recognition of this recovery and of their cultural, historical, and economic importance, the United States passed legislation that designates the native bison a national animal symbol [9].

Of course, there are inherent dangers to implementing a royalty as policy for iconic species. It would be an own goal if British businesses, corporations, and institutions began avoiding using the national animal symbol altogether if doing so were to incur a certain responsibility. Furthermore, for a cultural conscience to successfully take hold within a given nation′s economy, government incentives or disincentives to participate would have to be put in place, as well as widespread efforts to garner the mass consumer support that an iconic national species could theoretically accomplish.

But the alternative, imminent reality is a world in which the lion symbol represents a species widely rendered as no more than a memory. If Britain took this on, it could not only become a global leader by making its non-native national animal a conservation priority and giving it a Lion′s Share of all that revenue it helps to generate, but also propel the country from its miserly 123rd placing in the global rankings in the international efforts for conservation [10].

## 3. Lucked Out

Unlike threatened wildlife, the world′s biggest sports stars rarely need worry about their pay cheque. The combined global seasonal revenue for the world′s top ten wealthiest sports leagues is £64 billion [11]. Yet almost three-quarters of the extant animal mascots parading the stadiums of the world′s wealthiest sports leagues are at least near threatened, and many critically endangered (Figure 2).

An innovative idea was proposed by senior Brazilian scientists when the vulnerable-listed three-banded armadillo, *Tolypeutes tricinctus*, was chosen as the mascot for the Football World Cup in 2014. These scientists called on FIFA and the Brazilian government to protect 1000 hectares of the Caatinga forest home of the mascot for every goal scored during the tournament. In conjunction, a conservation plan for the species itself was also proposed [12]. The armoured mammal rapidly became the most successful FIFA World Cup mascot of all time, generating millions of dollars in revenue through merchandising [13]. But after the excitement that centred around the mascot dissipated, FIFA did not support the proposed environmental initiatives. It would only have cost a fraction of the event’s revenue to support the initiative. It would also have looked good for FIFA [14].

The idea of species royalties is not a new one. But the example of FIFA is a grave reminder that while lucrative individual battles continue to be more often lost than won, government policy to ensure royalty payments in exchange for the symbolic, cultural use of endangered species would eradicate missed opportunities such as this. Moreover, programmes such as the University of Missouri-Columbia′s 1999 *Mizzou Tigers for Tigers* conservation campaign for its Bengal tiger mascot [15], or IUCN′s *SOS* “Save your logo!” fund directed toward the private sector, demonstrate that it is not ideas, objectives, or even successful outcomes that are lacking. The challenge is finding a way to administer them on an international scale to contend with the sixth mass extinction that the planet is currently experiencing.

Following decades of intensive scientific endeavour, the toolkit to address wildlife conservation is well stocked, but without funds its use is constrained. Existing approaches to conservation are struggling to contend with the rapid rate of this mass decline. To confront this, the approach to conservation must now evolve quickly across disciplines, continents, cultures and economies [16].

The potential of the sporting market economy does not end with the teams′ individual revenues. High-profile sports events and personalities could also contribute towards endangered mascot awareness. Think Super Bowl half time commercials, the NBA′s 24.8 million Twitter followers, and Cristiano Ronaldo′s 215 million “friends” across social media. When Ronaldo posted 255 times on social media promoting a brand during 2016, he generated £137.4 million in media value [17]. Putting a familiar face on the biodiversity problem, as well as taking advantage of people’s intense attachment to their teams and mascots, could stimulate donations from those who might not normally contribute to conservation. And with millions of fans for each league around the globe, the pool of potential donors is immense.

## 4. Dressed to Conserve

Contemporary consumers have proven to be receptive to responsible brands, which have now overtaken “conventional” brands in terms of growth rate, according to IRI and Boston Consulting Group′s 2015 European study. And just as luxury brands have the power of transforming an ordinary self into a sophisticated and glamorous one, a recent research study illustrates how “the luxury of performing significant acts of altruism holds the power to transform an ordinary self into a desired self” [18].

This suggests that paying a species royalty for fashion that uses either an animal′s image or print could be an appealing prospect for the 21st century consumer. If such a policy were implemented, wearing endangered animal symbols could be the mark of having contributed towards that species′ conservation. Looking good would never have never felt so good.

The patterns of the varied spotted big cats are the most often represented wild species in Western fashion (Figure 3). Synonymous with power, savagery, barbarism, and feminine otherness, this association originated in the huntress in ancient Greek mythology, where the leopard held appeal because the female of the species was thought to be the superior fighter [19]. In ancient Egypt, Tutankhamun′s tomb combined real skins with faux prints crafted from woven linen appliquéd with stars within circles to represent a leopard′s rosettes.

Napoleon′s archeological discoveries in Rome, Greece, and Egypt reignited interest in all things ancient and classical in 19th century Europe, and spurred a popularity in the use of animal prints in textiles [20]. With the simultaneous birth of haute couture in France, designers looked to the animal kingdom, and the bold, graphic patterns of wild animals have been used by designers ever since. Several centuries ago, however, leopard print was reserved for the modish trimmings and accessories of the fashionable elite. Today it saturates both high-street and high-fashion.

Away from the catwalk, rampant bushmeat-, fur-, and body-part-poaching, habitat destruction, and poorly managed trophy hunting have eliminated leopards from at least 40 percent of their historic range in Africa, and over 50 percent in Asia. But if a species royalty were paid for animal print, the leopard could be the cash-cow of the jungle, benefitting all wildlife that shares their impressive (albeit reduced) domain which, as reported by Macdonald et al. (submitted) makes them the most potent “ambassador species” of any surviving mammal.

Just as species themselves adapt to changing environments and ecosystems, the history of fashion’s appropriation of animal skins, prints, and symbolism is also a document of society’s changing attitudes and ambivalences towards human–animal relations. As fashion continues to clothe its clients in the patterns of giraffes, zebras, tigers, and snakes, could the next chapter of this relationship be a more mutually supportive one?

## 5. Conclusion—Keeping the Wolf from Death′s Door

Establishing a cultural species economy could go some way in reconciling symbolic affluence with the pauperized reality of the planet′s wildlife. Undoubtedly, the intricacy of the ethics of species commodification requires careful consideration. In proposing a species royalty scheme that will primarily benefit the planet′s most enigmatic species, many other less anthropomorphically “appealing” wildlife could be financially left out in the cold. But charismatic animal symbols such as the leopard and lion are also large apex predators. Therefore, all wildlife sharing their habitat would benefit from their financial reward [21].

On 2 July 2015, the killing of a lion nicknamed “Cecil” prompted the largest global reaction in the history of wildlife conservation [22]. What the fallout of this moment has shown, is that if the powerful sentiment felt by millions of citizens worldwide is grasped, it could fund a movement to repay the historic cultural debt to animals. To do this, it is necessary to take stock of the proliferation of animal symbols, prints, and logos that adorn clothes, food, branding, and buildings. For centuries, they have brought human civilisation’s feelings of luck and protection, helping shape personal, professional, and national identities. It is now our turn to protect them and their habitat. 

## Figures and Tables

**Figure 1 animals-07-00052-f001:**
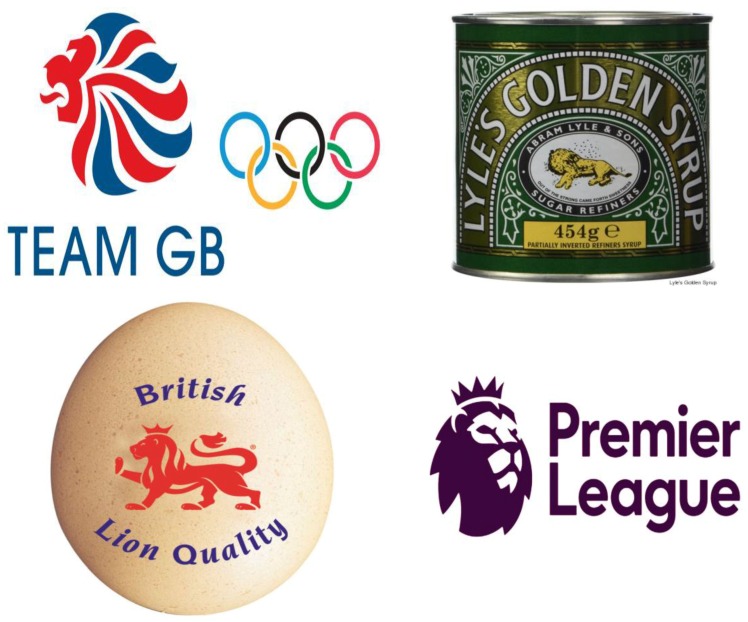
Top left: Logo of Great Britain′s Olypmic Team. Top right: Lyle′s Golden Syrup. Bottom left: British Lion Quality eggs. Bottom right: Logo of the Premier League.

**Figure 2 animals-07-00052-f002:**
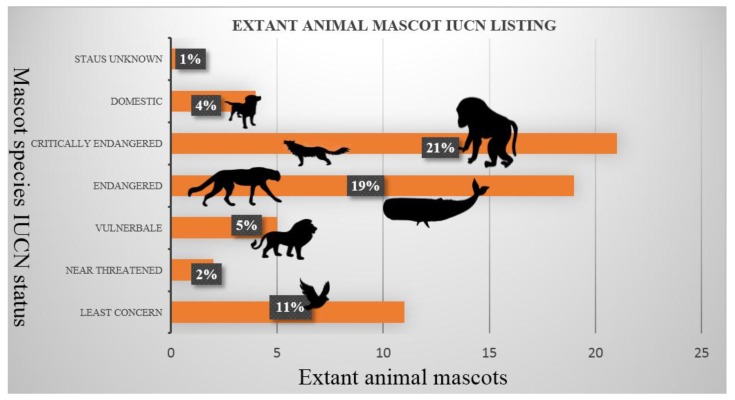
The extant animal mascots of the world′s top 10 wealthiest sports leagues.

**Figure 3 animals-07-00052-f003:**
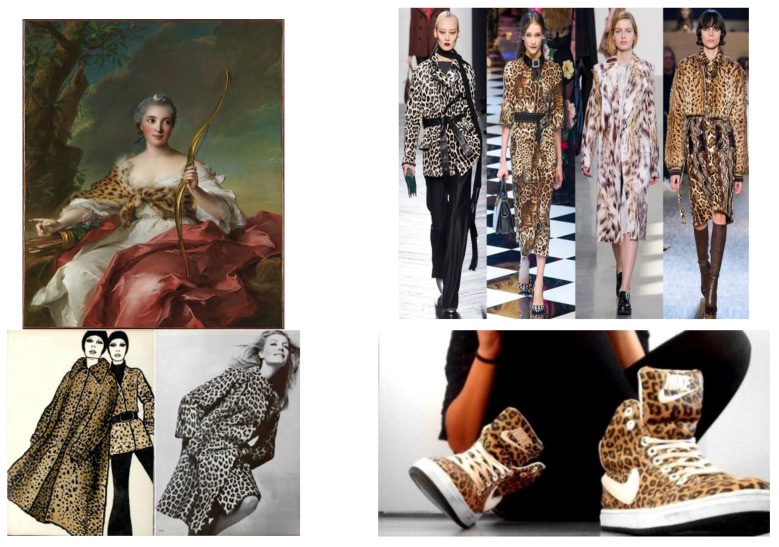
Top left: Madame de Maison-Rouge as Diana by Jean Marc Nattier. Met Museum, New York. Top right: examples of leopard print on fashion catwalks Autumn/Winter 2017. Bottom left: examples of vintage leopard coats. Bottom right: Nike shoes in leopard print.

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
