# Peer review of "A Cultural Conscience for Conservation"

_animals, 2017, doi:10.3390/ani7070052_

Round 1

Reviewer 1 Report

Dear authors,

For an Opinion Piece it seems important to me to make a distinction between 'Opinion' and 'Propaganda'. Where 'Opinion' keeps a critical stance towards the topic at hand, 'Propaganda', just tries to sell it, without clearly articulating possible critiques. If this is a useful distinction and framing, I think this Opinion Piece may be considered a bit more propaganda than opinion. I can see its merit, but would suggest that the authors use their combined intellectual resources to give the article (also) more of a critical twist, in order to live up more to the meaning of the word 'opinion'.

I can for instance imagine that various debates on 'commodification of nature / species' offers one avenue for a more intellectual critique on 'cultural consciousness'. Another opportunity for a more critical reflection could be to juxtapose the idea of 'cultural consciousness' with 'sustainable use' ('if it pays it stays') and see where it brings you.

Good luck and looking forward to the published version!   

Author Response

For an Opinion Piece it seems important to me to make a distinction between 'Opinion' and 'Propaganda'. Where 'Opinion' keeps a critical stance towards the topic at hand, 'Propaganda', just tries to sell it, without clearly articulating possible critiques. If this is a useful distinction and framing, I think this Opinion Piece may be considered a bit more propaganda than opinion. I can see its merit, but would suggest that the authors use their combined intellectual resources to give the article more of a critical twist, in order to live up more to the meaning of the word 'opinion'. I can, for instance imagine that various debates on 'commodification of nature/species' offers one avenue for a more intellectual critique on 'cultural consciousness'. Another opportunity for a more critical reflection could be to juxtapose the idea of 'cultural consciousness' with 'sustainable use' ('if it pays it stays') and see where it brings you. 

This is a fascinating point of definition. As an opinion piece it is our understanding we are to some degree presenting our own point of view, which could, and indeed should, be interpreted as its own form of argument and even advocacy (one might choose to use the word “propaganda” here, although we’re inclined not to because of its abundant baggage). At the same time as presenting our views (an opinion), it is our wish to present our opinion as an interesting, and even more importantly, a useful one. Therefore, much stimulated by the reviewer’s remarks we have revised our text to ensure that we do so in a measured and considered way. In light of the reviewer’s comments, we have taken steps at various points throughout the article to signpost the possible intricacies around the ethics of the commodification of wildlife. Specifically:

In lines 4346, we have made alterations that clarify the positioning of the piece as one considering the financial potential of a cultural conscience, rather than a propaganda piece that pushed the matter with blinkers and without rational argument for one thing.

In lines 99104, we have now provided further signposting to the potential pitfalls of these kinds of policies, as well as highlighting the lengths governments and NGO’s would have to go to achieve this kind of funding revolution.

In lines 194199, we have explored the issue of imbalance considering how this approach would by no means benefit all species.

As stated in the simple summary and abstract, the intention of this article is to ignite a multi-disciplinary exchange on the global cultural economy’s use of endangered species symbols. This piece is a call-to-action for research-, culture-, and business-communities to implement a new ethical phase in economic policy that recognises the global cultural debt to the world’s most charismatic wildlife species.

We believe we also make it clear that the proposed mechanism for raising conservation funds is only one of diverse mechanisms that will have to be brought to bear to raise the very large sums of money that will assuredly be necessary.

Reviewer 2 Report

Very interesting idea but I would suggest that it would be easier to apply moving forward rather than applying it retrospectively.  I would also suggest adding a few sentences on the "tiger in your tank" campaign by Esso in 1964 (now ExxonMobil).  Since 1995, ExxonMobil have been providing $1 million a year to The Tiger Fund (considerably less than the sums being proposed in the "comment").  I do not have the reference but there was a public opinion result in which Esso petrol was considered to be more eco-friendly than other brands.  So the company has, I suspect, reaped a considerable sales benefit. 

In fact, animals are very widely used in advertising.  There is an interesting article by SM Stone (The psychology of animals in advertising) from a Hawaii symposium in 2014 that might be useful to cite.  Also, have a look at the negative aspects on conservation of using animals in advertising (Schroepfer KK et al, 2011, Plos ONE (6(10): e26048).

Author Response

Comments and Suggestions from Reviewer 3 

Very interesting idea but I would suggest that it would be easier to apply moving forward rather than applying it retrospectively.

We have added a clarifying sentence in the introduction to address thiscertainly it was never the intention of this piece to argue for retrospective “taxation”. It was specifically intended as a future call-to-arms for nations and corporations to champion national animals as flagship conservation species where applicable. See lines 4647.

I would also suggest adding a few sentences on the "tiger in your tank" campaign by Esso in 1964 (now ExxonMobil). Since 1995, ExxonMobil have been providing $1 million a year to The Tiger Fund (considerably less than the sums being proposed in the "comment"). I do not have the reference but there was a public opinion result in which Esso petrol was considered to be more eco-friendly than other brands. So the company has, I suspect, reaped a considerable sales benefit. 

This seemed most relevant in relation to the consideration already given in the article to how beneficial this kind of association could be to a company’s public image. See note 14 to support note in line 130.

In fact, animals are very widely used in advertising. There is an interesting article by SM Stone (The psychology of animals in advertising) from a symposium held in Hawaii in 2014 that might be useful to cite. Also, have a look at the negative aspects on conservation of using animals in advertising (Schroepfer KK et al, 2011, Plos ONE (6(10): e26048). 

These are certainly interesting avenues of discussion that could well emerge from this piece in the form of another article.

Reviewer 3 Report

Overall comment

This paper presents an interesting new perspective on the perennial and increasingly urgent question about how we fund species conservation at a level to be effective and sustainable. It does, however, need to give at least an initial idea of how this might work. IUCN’s SOS fund started by appealling to companies using animal logos to donate funds -- “Save your logo!”. It totally failed to gain any traction with companies. How would the proposed royalty system work -- through legislation? By widespread consumer awareness campaigns? Companies would need either incentives or disincentives to participate (they don’t give away money unnecessarily) -- whether that’s in the form of legal requirements, or consumer support, or both? Without some indication of how this might work, the paper is a neat idea, but not likely to gain significant traction.

Specific comments

Line 18: Is this true? Or are the authors more aware of it because they were directly studying Cecil? Also major outcries for other individuals killed, e.g., the elephant Satau. A more general point is that individuals tend to generate more public outcry than even huge swathes of a population of a species.

Line 48 onwards, Section 2: Would be good to say upfront that this section focuses on UK brands. That’s in contrast to the next section which is global.

Lines 74-75: Some further detail is needed here. How much money would be raised? What is a lion guardian?

Line 99: What does successful mean in this context?

Lines 109-115: This is somewhat of a non-sequiteur. The previous argument has all been about royalties from brands and logo use. Individual donors are a different issue. Clearly important, but a different argument, unless linked back to their teams’ logos.

Lines 117-119: Just in Europe? Assume not in other parts of the world, e.g., Asia with some of the world’s largest and most rapidly-growing firms?

Author Response

Overall Comment 

This paper presents an interesting new perspective on the perennial and increasingly urgent question to how we fund species conservation at a level to be effective and sustainable. It does, however, need to give at least an initial idea of how this might work.

We have now addressed how this might work in lines 8898. We have gone about this by including two past examples of highly successful cases that could help to provide helpful case studies for future action.

IUCN’s SOS fund started by appealing to companies using animal logos to donate funds“Save your logo!”. It totally failed to gain any traction with companies. How would the proposed royalty system workthrough legislation? By widespread consumer awareness campaigns?

This is a very useful example. It has been added into the article to compliment the previous stand alone example of the University of Missouri-Columbia’s 1999 Mizzou Tigers for Tigers conservation campaign. These kind of examples provide important reminders that attempts have been made to implement similar schemes in the past that have not provided a large-scale revolution

Companies would need either incentives or disincentives to participate (they don’t give away money unnecessarily)whether that’s in the form of legal requirements, or consumer support, or both? Without some indication of how this might work, the paper is a neat idea, but not likely to gain significant traction.

Specific Comments 

Line 18: Is this true? Or are the authors more aware of it because they were directly studying Cecil? Also major outcries for other individuals killed, e.g., the elephant Satau. A more general point is that individuals tend to generate more public outcry than even huge swathes of a population of a species. 

Yes true, journal article source is cited in reference 23.

Line 48 onwards, Section 2: Would be good to say upfront that this section focuses on UK brands. That’s in contrast to the next section which is global. 

Now clarified in line 45.

Lines 7475: Some further detail is needed here. How much money would be raised? What is a lion guardian? 

Now detailed in reference 5.

Line 99: What does "successful" mean in this context? 

More detail added to clarify in line 127128.

Lines 109115: This is somewhat of a non-sequitur. The previous argument has all been about royalties from brands and logo use. Individual donors are a different issue. Clearly important, but a different argument, unless linked back to their teams’ logos. 

We have adapted and now linked back to mascot subject, see lines 144146.

Lines 117119: Just in Europe? Assume not in other parts of the world, e.g., Asia with some of the world’s largest and most rapidly-growing firms? 

The study referenced here focuses on Europe, hence our focus on Europe in the comment.